# A Comparative Study of Biogas Production from Cattle Slaughterhouse Wastewater Using Conventional and Modified Upflow Anaerobic Sludge Blanket (UASB) Reactors

**DOI:** 10.3390/ijerph17010283

**Published:** 2019-12-31

**Authors:** Mohammed Ali Musa, Syazwani Idrus, Mohd Razif Harun, Tuan Farhana Tuan Mohd Marzuki, Abdul Malek Abdul Wahab

**Affiliations:** 1Department of Civil Engineering, Faculty of Engineering, University Putra Malaysia, Serdang 43400, Malaysia; alisulezee@gmail.com (M.A.M.); t_nurfarhana@yahoo.com (T.F.T.M.M.); 2Department of Civil and Water Resources Engineering, University of Maiduguri, Maiduguri P.M.B. 1069, Nigeria; 3Department of Chemical and Environmental Engineering, Faculty of Engineering, Universiti Putra Malaysia, Serdang 43400, Malaysia; mh_razif@upm.edu.my; 4Faculty of Mechanical Engineering, Universiti Teknologi Mara, Shah Alam 40450, Malaysia; abdmalek@uitm.edu.my

**Keywords:** anaerobic digestion, cattle slaughterhouse wastewater, UASB reactors, COD, HRT, OLR

## Abstract

Cattle slaughterhouses generate wastewater that is rich in organic contaminant and nutrients, which is considered as high strength wastewater with a high potential for energy recovery. Work was undertaken to evaluate the efficiency of the 12 L laboratory scale conventional and a modified upflow anaerobic sludge blanket (UASB) reactors (conventional, R1 and modified, R2), for treatment of cattle slaughterhouse wastewater (CSWW) under mesophilic condition (35 ± 1 °C). Both reactors were acclimated with synthetic wastewater for 30 days, then continuous study with real CSWW proceeds. The reactors were subjected to the same loading condition of OLR, starting from 1.75, 3, 5 10, 14, and 16 g L^−1^d^−1^, corresponding to 3.5, 6, 10, 20, 28, and 32 g COD/L at constant hydraulic retention time (HRT) of 24 h. The performance of the R1 reactor drastically dropped at OLR 10 g L^−1^d^−1^, and this significantly affected the subsequent stages. The steady-state performance of the R2 reactor under the same loading condition as the R1 reactor revealed a high COD removal efficiency of 94% and biogas and methane productions were 27 L/d and 89%. The SMP was 0.21 LCH_4_/gCOD added, whereas the NH_3_-N alkalinity ratio stood at 651 mg/L and 0.2. SEM showed that the R2 reactor was dominated by Methanosarcina bacterial species, while the R1 reactor revealed a disturb sludge with insufficient microbial biomass.

## 1. Introduction

Recently, climate change has become a global phenomenon with far-reaching consequences on the environment leading to an increase in renewable energy demand at both the national and international levels. The use of excessive fossil fuel, improper management of biodegradable solid and liquid waste, and agricultural activity [1] are presumed to be the leading cause of environmental deterioration [2,3,4]. For instance, an increase in the amount of carbon dioxide (CO_2_) and methane (CH_4_) in the atmosphere, known as greenhouse gases, constitutes the leading cause of global warming. A reasonable proportion of these greenhouse gas emissions are related to the uncontrolled degradation of organic matter contained in the increasing amount of human-produced waste [5,6,7]. Cattle slaughterhouses are indirectly a product of agricultural activity (livestock farming) and are among the many food industries that are utilizing a large quantity of freshwater and generate a considerable volume of wastewater which is rich in organic contaminants and nutrients [8,9]. Improperly treated and discharged wastewater generated from slaughterhouses constitutes a potential threat to public health and the environment [10], due to the high amount of potentially hazardous organic and potentially pathogenic microorganism. However, the quantity of wastewater generated from slaughterhouses varies, depending on the number of animals and the cleaning technique employed. According to Yung et al. [11], more than 65% of the freshwater used in slaughterhouses is associated with spraying, rinsing, and cleaning activities. The remaining 35% of the water use is attributed to cooling water, scald tanks, animal handling facilities, tools sterilization, vehicle washing, and personal hygiene. In another development, the amount of water consumed per animal slaughtered is between 1.0 and 8.3 m^3^ [12], and 0.4 to 3.1 m^3^, as reported by Claudia et al. [13]. The wastewater generated from slaughterhouses, during and after the operation, is considered as high strength wastewater due to a high level of chemical oxygen demand (COD); fats, oils and grease (FOG); nitrogen and phosphorus; total suspended solids (TSS); and colloidal compounds such as blood, protein, and cellulose [14]. However, the strength may differ from one industry to another depending on the number and type of the animal’s slaughter, as well as the operational processes involved. 

Slaughterhouse wastewater (SWW) effluent may contain COD concentrations between 3000 to 30,000 mg/L [14,15]. Rajab et al. [16] reported a SWW contamination concentration with COD (3102 ± 688 mg L^−1^); fats, oils and grease (FOG) (375 ± 151 mg L^−1^); suspended solids (SS) (872 ± 178 mg L^−1^); total Kjeldahl nitrogen (TKN) (186 ± 27 mg (N) L^−1^); and total phosphate (PO_4_^−3^-P) (76 ± 36 mg L^−1^). Moreover, the work of Johns and Harrison [17] also revealed the COD concentrations of two different slaughterhouses varying from the medium of (1000 to 3000 mg L^−1^) to high a strength of (5000 to 10,000 mg L^−1^). The discharge of untreated high nitrogenous and colored effluent, such as slaughterhouse wastewater, could promote algal bloom formation and render the bottom of the receiving water body anaerobic, thereby caused eutrophication, i.e., a situation whereby light penetration to the aquatic environment is inhibited and dissolve oxygen supply is depleted leading to the death of aquatic animals and the release of methane gas into the atmosphere [15,18]. Conventionally, SWW treatment technologies are similar to those applied in municipal wastewater treatment. It includes chemical coagulation, lagoon pond systems, sedimentation and filtration, adsorption, chlorination, and high rate oxidation ditches [14]. These methods produce high-quality effluent, however, the large area space requirement for plant installation, huge sludge production, high energy demand, offensive odor, lack of energy recovery, and the overall cost of operation and maintenance have been reported as the major drawback [19]. Due to pressure from environmentally aware consumers, various guidelines and water quality criteria have been set by both local and international committees. For example, the Department of Environment Malaysia (DOE) have set standards A and B for all intending investors capable of generating waste as sewage into the receiving water body [20]. Similar standards for treated effluent being discharged into a receiving catchment in eThekwini Municipality (South Africa) have been reported by Naidoo and Olaniran [21]. These standards are now driving industries generating high strength wastewater to develop new purification technology capable of meeting the standard discharge regulations [22]. 

Anaerobic digestion technology has appeared to be a promising technique due to its efficiency in waste volume reduction while producing energy in the form of biogas and digestate as biofertilizer [23]. The anaerobic digestion process involves four groups of bacteria responsible for degrading the complex organic matter, however, the conversion process of the complex substrate to biogas is a function of the mutually inclusive symbiotic relationship among the microorganism. These include the fermentative bacteria, syntrophic acetogens, homoacetogens, hydrogenotrophic methanogens, and acetoclastic methanogens [24]. Typical biogas consists mainly of 66.1% CH_4_ and 33.3% CO_2_ and it also contains some trace compounds like N_2_ (0.5) and O_2_ (0.1) [25], however, several factors determine a biogas plant’s successful activity. From an environmental and economic point of view, the feedstock of substrates and their transport is a very important issue, although improving these processes determines the benefits that the waste-to-energy system can offer [26].

Anaerobic reactors such as continuous stirred tank reactor (CSTR), upflow anaerobic sludge blanket reactor (UASB), anaerobic fluidized bed reactor (AFBR), anaerobic rotating biological contactors (RBC), anaerobic fixed-film bioreactor (FFB), and expanded granular sludge bed (EGSB) have been employed for treatment of wastewater [27]. Among these, the UASB is mostly preferred and widely adopted anaerobic option for the treatment of various wastewater such as starch wastewater [28], cheese whey [29], potato juice [30], and palm oil mill effluent [31], which could be due to its high flexibility, high efficiency, biogas production, and overall lower cost of maintenance. On the one hand, for example, a study of morphology response and microbial community structure of influent COD/SO_4_^2−^ ratio in UASB reactor by Lu et al. [32] revealed a biogas production of 1.15 ± 0.21 L L^−1^d^−1^, 1.16 ± 0.22 L L^−1^d^−1^, and 1.17 ± 0.33 L L^−1^d^−1^ at COD loading rates of 10, 5, and 3 g/L, whereas the corresponding methane yields were 67.3% ± 3.1%, 64.0% ± 1.7%, and 61.3% ± 1.6%, respectively. In their previous study, a biogas production of 1.15 ± 1.17 L L^−1^d^−1^ was recorded [33]. On the other hand, Claudia et al. [13] evaluated a three-phase separation system UASB reactor treating slaughterhouse wastewater and found a biogas production of 11 L^−d^ (STP). The COD removal efficiency varied between 77% to 91% at an average OLR of 3.5 and 6.5 kg COD·m^−3^d^−1^. However, numerous challenges are associated with UASB technology, especially in the treatment of slaughterhouse wastewater treatment. These include slow-growing microorganisms, sludge washed out during effluent discharge, reduced tolerance to fats, oils and grease, scum formation on the substrate surface within the reactor, and a long startup period [34]. Thus, these could affect sufficient biogas formation in the reactor. Therefore, the objective of this study is to evaluate the performance of a conventional (R1) reactor and a modified (R2) UASB reactor treating cattle slaughterhouse wastewater (CSWW) and to use the most effective result obtained to serve as the basis for improving bioreactor design in cattle slaughterhouses. 

## 2. Materials and Methods 

### 2.1. Waste Characterization

Cattle slaughterhouse wastewater (CSWW) was collected from the discharge point of Selangor abattoir, Malaysia. The abattoir is the biggest in the whole state of Selangor. The sample was immediately transported to the public Health and Environmental Engineering Laboratory at the Universiti Putra Malaysia. Upon arrival, the CSWW was screened to remove larger particles and subsequently analyzed for physical and chemical parameters using standard methods of water and wastewater examination method (2450-G) [35]. After characterizing the sample, it was then kept at 4 °C until further use. The physicochemical analysis of the raw sample is presented in Table 1.

### 2.2. Reactor Design

Two identical conventional (R1) and modified (R2) upflow anaerobic sludge blanket (UASB) reactors (Figure 1) were used for the experiment. Both reactors were made from polyvinyl chloride (PVC) tubes. The reactors consisted of an internal diameter of 180 mm and a height of 700 mm. Each of the reactors had a working volume of 12 L. At the top of each reactor, a gas sample bag of 6 L was connected to the gas outlet tube of the two reactors. Apart from the influent inlet at the bottom and gas outlet flows on the top, each reactor had an equally spaced effluent outlet at a height of 210 cm from the bottom. The heating element was placed in between the two reactors with an immersed edge in the water bath, and a thermometer was also immersed to monitor the temperature of the water bath. The R2 reactor was packed with 25 mm synthetic grass (SG) (BLS HOME DECO 25 mm 1 m × 1 m at the bottom-top and round cylindrical shape at 200 mm height from the bottom. The SG media was used as an attached growth surface. The top flat-round perforated PVC was also coupled with SG to provide better settling and reduce sludge washout. Both reactors were operated at a mesophilic temperature of (35 °C ± 1 °C), with 85% of the reactor height submerged in a water bath. The reactors were run in upflow mode. Feeding was done using a peristaltic pump from the bottom through a blanket of biologically activated sludge at a constant flow rate to provide sufficient contact between the cattle slaughterhouse wastewater (CSWW) and the seed sludge containing microbes.

### 2.3. Inoculum and Synthetic Wastewater

The two reactors were inoculated with seed sludge collected from the municipal wastewater treatment plant at the Faculty of Engineering, Universiti Putra Malaysia. Each reactor was loaded with 6 L of seed sludge, occupying about 45% of the total reactor volume from the bottom. The TSS, pH, and COD of the seed sludge are 12,410 ± 125 mg/L, 7.8 ± 0.3, and 15 g/L. Before the continuous study, all the reactors were acclimatized with synthetic wastewater prepared according to Rosli et al. [36]. The composition of the synthetic wastewater is as follows: urea, 2.14 g; yeast, 23 g; dried blood, 5.75 g; full cream milk, 144 mL; ammonium phosphate (NH_4_)_2_HPO_4_, 3.4 g and top up to 1 liter with tap water. The acclimatization period of the R1 reactor lasted for four weeks and three weeks for the R2 reactor.

### 2.4. Analytical Methods

The effluent from the two bioreactors was sampled at a one-day interval throughout the experimental period. It was then analyzed for COD, ammonia nitrogen NH_3_-N, alkalinity, and volatile fatty acids VFAs. The pH of the systems was analyzed using (Mettler-Toledo AG, 8603 Schwarzenbach, Switzerland), while COD was determined according to standard methods for the examination of water and wastewater (2450-G) [35]. Spectrophotometer (HATCH DR 900, Agilent, Santa Clara, CA 95051, USA) (Salicylate powder pillow method 8155) was used for measuring ammonia nitrogen (NH_3_-N). Alkalinity was determined using the titrimetric method, and the samples were evaluated as partial alkalinity (PA), by titration to pH 5.75 and total alkalinity (TA) by titration to pH 4.3 with 0.02N H_2_SO_4_ using 50 mL burette (DIN AS Germany). VFAs samples (Effluent) analysis was carried out using gas chromatograph (HP 6890 N) employing a 100 m × 0.25 mm ID × 0.2 µm Supelco SP 2560 capillary column (supelco, Inc., Bellefonte, PA, USA). The carrier gas was nitrogen, and the flow rate was 1.2 mL/min. Furthermore, 1 µL of fatty acid methyl esters (FAME) was injected using an auto sample into the GC, which was equipped with a flame ionization detector (FID), at a split ratio of 1:20. The injector and detector temperatures were maintained at 250 °C and 270 °C, respectively. In addition, the biogas composition analyses were carried out using a gas chromatograph (Agilent HP 6890 N) equipped with a thermal conductivity detector (TCD). The column was HP Molesieve (Agilent Technologies, Santa Clara, CA, USA) of 30 m length × 0.5 mm ID × 40 μm film thickness capillary column. The splitless inlet, oven, and TCD detector temperatures were all kept at 60 °C, 70 °C, and 200 °C, respectively. Argon functioned as the carrier gas, while nitrogen was used as the makeup gas. Scanning electron microscopy (SEM) analysis was carried out to examine the sludge and the packing material surface morphology following the osmotic dehydration method with ethanol. 

### 2.5. Startup of Conventional and the Modified UASB Reactors

In the previous studies of wastewater treatment using anaerobic digestion process, the startup of the reactor is done by either increasing OLR while reducing HRT or increasing OLR corresponding to an increase in COD concentration while maintaining constant HRT [37,38]. In this study, the latter strategy was employed by maintaining the same HRT to avoid unstable operation and gradually increasing OLR. Table 2 summarizes the strategies employed during the startup period of the two UASB reactors. 

## 3. Results and Discussion

### 3.1. Biogas, Methane, and Specific Methane Productions

A steady-state condition in anaerobic digestion is attained when the biogas and the corresponding methane production remained unchanged in volume and composition or with little or negligible differences, i.e., it is assumed a steady-state condition is reached. Marcos et al. [39] showed that a steady-state condition is achieved when the methane production remained constant consecutively. Figure 2, Figure 3 and Figure 4 reveal the amount of biogas, percentage of methane, and the specific methane production during the whole period of comparison between the R1 and R2 UASB reactors that lasted for 95 days. A change in the influent COD concentration corresponding to change in OLR was carried out from one stage to another once the biogas production attained a steady-state condition with less than 5% variation. The initial OLR of 1.75 g L^−1^d^−1^ was gradually elevated until 16 g L^−1^d^−1^ OLR was introduced to both reactors. As shown in Figure 2, at a stable state OLR 1.75, 3, and 5 g L^−1^d^−1^, the biogas productions in the R1 and R2 UASB reactors were 6.2, 7.9, and 10.2 L and 6.8, 8.6, and 12.2 L, respectively, and these corresponded to methane concentrations of 71%, 65%, and 75% for the R1 reactor and 92%, 85%, and 85% for the R2 UASB reactor (Figure 3). 

Similarly, the corresponding specific methane production within these OLRs were found to be 0.2, 0.14, and 0.12 LCH_4_/g COD_added_ in the R1 reactor and 0.26, 0.2, and 0.19 LCH_4_/g COD_added_ in the R2 UASB reactor (Figure 4). A further increase in OLR by feeding the CSWW to 10 g L^−1^d^−1^ led to an increase in biogas production that reached approximately 27 L d^−1^ at a steady state in the R2 reactor (Figure 2), whereas a drastic reduction of biogas production was recorded in the R1 reactor under the same condition of loading. The stage (10 g L^−1^d^−1^) in which the high volume of biogas was produced corresponds to 20 g L^−1^ COD_added_, and the methane concentrations were 89% and 45% in the R2 reactor, while the specific methane production (Figure 4) was 0.21 and 0.03 LCH_4_/g COD_added_, respectively. Subsequently, the OLR was then increased to 14 g L^−1^d^−1^, after which the biogas and methane concentrations for the R1 and R2 UASB reactors decreased to 5.8 and 22 L, corresponding to 0.011 and 0.084 LCH_4_/g COD_added_ (Figure 4). It is obvious that the drastic decrease in the biogas and methane concentrations of biogas decreases with an increase in OLR. Moreover, the rate of biogas, methane concentration, and the specific methane production continued to decrease with an increase in OLR to 16 g L^−1^d^−1^ (Figure 2, Figure 3 and Figure 4), and this was attributed to the shock received by the systems due to changes in the OLR that affect the methanogenic activities especially in the R1 reactor. However, the decrease was quite severe in the R1 reactor as compared to the R2 UASB reactor. Therefore, increasing the OLR to 10 g L^−1^d^−1^, resulted in an optimum specific methane yield of 0.21 LCH_4_/g COD_added_.

Several studies on anaerobic digestion have shown that biogas production and COD removal can be more efficient with changing feed strategies. For instance, the research of Basitere et al. [40] demonstrated that biogas production was improved when feedstock was fed more often than once per day. Similarly, the methanogens were not inhibited throughout the study period of the R2 reactor as both the levels of VFAs, as well as the pH, were stable. However, the massive decline in the performance of the R1 reactor when the OLR was increased to 10 g L^−1^d^−1^ was due to the shock received by the microbial population beyond the degradation capacity of the biomass. The problems encountered in the R1 reactor could also be attributed to the high scum formation and the shock loading, thus, resulted in the loss of the methanogenic population. The investigation of Borja et al. [41] showed that methane content of biogas produced during anaerobic digestion of slaughterhouse wastewater decreases from 78% to 57% when the OLR was increased from 2.9 to 54 g COD L^−d^. The decrease in methane content was due to inhibition of the methanogenic bacteria at a higher loading rate, which caused an increase in effluent VFA contents. Another important inhibitor that could be attributed to the poor performance of the R1 reactor was the significant presence of ammonia and lack of trace elements. Furthermore, the microorganisms responsible for the bioconversion of substrate to produce biogas in the R1 reactor may also be inhibited by a plethora of substances and products which may enter the process from an external source or may be generated by the process. Consequently, the behavior of the R1 reactor from OLR 10 to 16 g L^−1^d^−1^ is an indication of strong inhibition of hydrolytic and acidogenic microorganisms, which subsequently affects the activities of methanogens following the sequence of degradation.

### 3.2. COD Removal Efficiency of the R1 and R2 Reactors 

The performance of conventional (R1) and the modified (R2) UASB reactors were operated continuously for 95 days and evaluated in terms of COD removal at various OLR, as shown in Figure 5. Both reactors were maintained at 24 h HRT and 36 ± 1 °C. The variation of OLR during the whole period of the study ranged from 1.75 g L^−1^d^−1^ to 16 g L^−1^d^−1,^ and this corresponds to an influent COD range of 3.5 g COD L^−1^ to 32 g COD L^−1^. The removal efficiencies of the R1 and R2 reactors at steady-state OLR of 5 g L^−1^d^−1^ were >90% on the average. However, with an increase in OLR to 10 g L^−1^d^−1^, the R1 reactor suffered a severe decline of COD removal to 48%, whereas the R2 reactor maintained a significant COD removal of 95% at a stable state. The setback observed in the R1 reactor could be due to shock received by the microbial community to withstand the change in loading rate. Moreover, the formation scum coupled with insufficient bacterial population resulted in a decline in the microbial activity of the R1 reactor as compared with the R2 reactor. In addition, the synthetic grass bed in the R2 reactor was able to provide sufficient room for the microbial population to multiply. The subsequent increase in OLR to both reactors further deteriorated the rate of COD removal efficiency in the R1 reactor to an average of 45% and 43% at OLR 14 and 16 g L^−1^d^−1^, whereas the R2 reactor was found to be 72% and 68%. The resilience of the R2 reactor was ascribed to the high adaptability of the microbial community to the environment [42]. Furthermore, the studies of Fang and coworkers had shown that the UASB reactor could maintain a stable process once the operation is within the normal OLR boundaries, which ranged between 1.5 and 16.0 kg COD m^−3^d^−1^ [28,30]. Consequently, the evaluation of the two experimental results indicates that the modified UASB reactor (R2) showed excellent efficiency at reducing COD as compared with a conventional UASB reactor. Similar wastewater treatment was previously reported in the literature [43] using a UASB reactor. However, the results obtained were lower as compared with the modified reactor in this study. In general, an increase in OLR and maintaining 24 h HRT for both reactors negatively affected the R1 reactor, whereas the R2 reactor was able to withstand high OLR with quality effluent at HRT of one day.

### 3.3. Variation of the Ammonia Nitrogen (NH_3_-N) Concentration in the R1 and R2 Reactors

Figure 6 depicts the profile of ammonia nitrogen concentration in the effluent of the R1 and R2 reactors treating cattle slaughterhouse wastewater at various OLR after the anaerobic digestion, which normally occurs as a result of protein hydrolysis [44]. There was no clear difference between the ammonia nitrogen produced between OLR 1.75 and 3 g L^−1^d^−1^ in both R1 and R2 reactors. A slight difference in NH_3_-N concentration was observed when the OLR was increased to 5 g L^−1^d^−1^, the values were from 347 mg/L to 481 mg/L and 54 mg/L to 631 mg/L in the R1 and R2 reactors, respectively. However, a significant difference was seen at OLR 10 g L^−1^d^−1^ in which the values of NH_3_-N decreased from 481 mg/L to 178 mg/L in the R1 reactor. These differences in the R1 reactor indicate that protein hydrolysis was not proceeding, while assimilation of nitrogen to microbial biomass was not occurring, depicting that the growth of microorganisms (acetogens and methanogens) was inhibited. In the R2 reactor, ammonia nitrogen values increased from 546 mg/L to 672 mg/L under the same OLR. The values of NH_3_-N in the R2 reactor could be related to the fast assimilation of nitrogen by the large population of microbial biomass. Comparing the performances of the R1 and R2 reactors at steady states showed that under stable operation, at low organic loading rate (1.75–5 g L^−1^d^−1^), both reactors were able to assimilate the protein content of the wastewater which indicated that the reactors were more efficient in converting the organic matter to methane at lower OLR, however, by increasing the organic loading rate from 10 to 16 g L^−1^d^−1^, the ammonia nitrogen level in the effluent of the R2 reactor was observed increase, corresponding to the increase in COD levels and organic loading rates. Although, Burke et al. [45] reported that the release of ammonia from the proteins triggered an increase in alkalinity in the system, which in turn led to increased pH values unsuitable for the development of methanogenic bacteria. Interestingly, the level of ammonia increase, in the effluent of the R2 reactor, neither inhibits nor caused any significant changes in the reactor operation. The absence of inhibition signs in the R2 reactor may be attributed to the higher activity of mesophiles, which were primarily responsible for converting the proteins to CH_4_ gas during the degradation process. The average values of NH_3_-N during the steady-state OLR, throughout the operational period of the R2 reactor, were below the toxic limits reported in the literature [46].

### 3.4. Variation in Alkalinity Ratio and pH Profile of the R1 and R2 Reactors

The high protein and lipid content of slaughterhouse wastewater makes it a challenging material for anaerobic digestion due to the inhibition caused during degradation. The functional anaerobic digestion system depends on the buffering capacity and the degree of adaptation of the microorganisms [47]. Alkalinity is an important parameter in anaerobic digestion that shows the capability of a solution to withstand a drop in pH produced by the release of organic acids, namely, the system buffer capacity [48]. The ability of methanogenic bacteria to resist higher VFA accumulation is highly dependent on the alkalinity value of the system, as well as its buffering capacity. Thus, the methane production processes in a bioreactor are a function of alkalinity and pH stability [49]. The alkalinity ratio has to be maintained in the range of (0.1–0.3). When the reactor partial and total alkalinity ratio exceeds 0.4, the system may attain unstable condition [36]. In this study, the alkalinity ratio of the R1 reactor increased gradually from 0.16 to 0.18 and 0.24, whereas for the R2 reactor they were 0.09, 0.19, and 0.17 at OLR 1.75, 3, and 5 g L^−1^d^−1^ (Figure 7). These values portray a stable operating condition and sufficient alkalinity, well below the inhibitory level. Subsequently, an increase in COD concentration to 20 g L^−1^ corresponding to 10 g L^−1^d^−1^ resulted in a rise in alkalinity ratio of the R1 reactor, quite above the normal working range. These findings indicated that feedstock dilution affected the stability and performance of the R1 reactor. This is probably due to a change in the buffering capacity, as shown by an increase in the alkalinity ratio to 0.8 (Figure 7). These affected the performance of the R1 reactor in the subsequent stages of OLR (14 and 16 g L^−1^d^−1^). The IA/PA ratio in the R1 reactor was relatively high and continuously increased, reaching a value above 1.4 at the end of the experiment. Therefore, this is an indication of process imbalance. Consequently, the alkalinity ratio of the R1 reactor significantly exceeded the optimal limit of 0.3, and this coincides with the research of Mata-Alvarez [50]. However, the alkalinity ratio at all the loading rates of the R2 reactor were less than 0.3, and the total VFA concentration was below 100 mg/L.

In anaerobic digestion, microorganisms have a working range of pH. Methanogens are sensitive to a pH between 6.5 and 7.5 and have an optimal pH of between 7.0 and 7.2 [51]. Whereas, the review of Morales-Polo et al. [52] showed that anaerobic bacteria need pH ranges for fermentation bacteria around 4 to 8.5, and methanogens between 6.5 and 7. Ugurlu and Forster [53] stated that their previous investigation had established variability in the optimal pH value, with the range depending on the substrate and digestion technique used. For efficient methane production, the anaerobic digestion process normally operates at the aforementioned optimal pH range. However, the formation of degradable intermediates (VFA) tends to lower the pH of the bioreactor during operation. Changes in the pH of bioreactors during operation, are easily detected through alkalinity (gCaCO_3_/L), which is the main indicator of the buffering capacity of the system. Figure 8 shows the effluent pH pattern of the R1 and R2 UASB reactors treating CSWW. The pH of the R1 reactor were around 6.8, 6.7, and 6.72 on average, whereas those for the R2 reactor were 6.92, 6.86, and 7.2 at OLR 1.75, 3, and 5 g L^−1^d^−1^, respectively. Subsequently, increasing the concentration of substrate COD to 10 g, L^−1^ d^−1^ resulted in a decrease in the pH of the R1 reactor to 4.2, while the R2 reactor maintained 7.3 at a steady state. The performance of the R1 reactor from OLR to 10 g L^−1^d^−1^ could be due to volatile fatty acids (VFAs) accumulation in the system beyond the microbial consumption. However, the R2 reactor depicts a system with a sufficient buffering capacity in which the VFAs were consumed by the microbial population. The performance of the R1 reactor further deteriorated, but still maintained pHs of 4.2 and 4.3 at 14 and 16 g L^−1^d^−1^ OLR.

### 3.5. Variation in VFA of the R1 and R2 Reactors

In wastewater treatment using anaerobic digestion, the characteristics and environmental parameters are very much important in the growth and maintenance of methanogenic bacteria. However, in anaerobic bioreactors, maintaining a sufficient number of methanogens is paramount to the overall performance of the system. Figure 9 demonstrates the volatile fatty acids concentrations in mg/L relative to time revealed during the comparative study of the R1 and R2 reactors. It was observed that the relatively low VFAs concentration in the R1 and R2 reactors between OLR 1.75, 3, and 5 L^−1^d^−1^, indicates that methanogenic activities are quite robust, and this was also reflected in specific methane production of the reactors (Figure 4). A rapid increase in acetic, propionic, and butyric acids was observed when the OLR was increased to 10 g L^−1^d^−1^ in the R1 reactor. However, there were no significant changes in acetic, propionic, and butyric acids observed in the R2 reactor under the same condition of loading, which implied that the experiment in these phases was highly stable. Furthermore, the performance of the R2 reactor revealed that the microbial consortium has efficiently adapted to the system. Surface scum formation was observed in the R1 reactor, thus, preventing biogas from flowing freely out of the reactor.

The poor performance of the R1 reactor could be attributed to the shock loading caused by the increasing feed concentration, particularly at 10 g L^−1^d^−1^ resulting in VFA accumulation. Ugurlu and C. F. Forster [53] also observed that the reactor under stress is usually due to an increase in feed concentration, which tends to accumulate VFA and reduce methane gas concentration. Similarly, at OLR 10, 14, and 16 g L^−1^d^−1^, all the organic matter was utterly acidified in the R1 reactor, which signifies a high dominance of acidogens and acetogens. Therefore, this implies that methanogens in the R1 reactor were unable to cope with the rate of organic matter conversion to biogas, hence the drastic loss of microbial population. Some studies have also shown that reactor failure is mainly due to the inhibition of methanogens and acetogens [54,55]. The inhibition process occurs when the long-chain fatty acids (LCFAs) disappear from the solution and accumulate in solid biomass within 24 h and are subsequently adsorbed into the membrane or cell wall of bacteria, which damages the microbial cell transport function or protective function [56]. Generally, VFAs produced in the anaerobic process could be ultimately transformed into CH_4_ and CO_2_ by syntrophic acetogens and methanogenic bacteria, however, VFAs can be accumulated at high organic loading, resulting in a decrease of pH and even the failure of the reactor [57,58].

### 3.6. Scanning Electron Microscopy (SEM) Analysis of Sludge in the R1 and R2 Reactors

Anaerobic digesters usually host a diverse methanogenic bacterial community, containing both acetoclastic and hydrogenotrophic methanogens, and thus enhancing reactor stability [59,60]. Figure 10a,b shows the image of sludge in the R1 reactor and the synthetic grass as attached growth with sludge in the R2 reactor after acclimatization, whereas Figure 10c,d represent the image of the sludge in the two reactors after contact with the CSWW for the whole period of comparison. The analysis of the sludge and the attached biomass growth on synthetic grass media were conducted using scanning electron microscopy. The observed morphologies in the R1 reactor (Figure 10a) and the R2 reactor (Figure 10b) were rod-shaped cells growing end-on-end in long filaments depicting the presence of *Methanosaeta* sp. The SEM analysis conducted in a study by Gomes et al. [61] support our findings. 

Figure 10d revealed coccoidal-shaped cells growing together in large aggregates, which suggest the presence of *Methanosarcina* sp. commonly found in anaerobic reactors at elevated OLR [62,63]. The population of the microbes in the R1 reactor is distinctly different from that of the R2 reactor, as can be seen in Figure 10c. This was presumably due to the presence of the attached growth media (synthetic grass in the R2 reactor). The shift in the microbial population growth favored *Methanosarcina* sp. with a change in operating conditions. Moreover, the synthetic grass media in the R2 reactor effectively supports the retention of high biomass growth (*Methanosarcina* sp.), and as such, the bacterial consortium was able to suppress the formation of high volatile fatty acids in the R2 reactor. Comparatively, the development of Methanosarcina species in the R2 reactor appears to be of critical importance. As compared to other methanogens in the R1 reactor (*Methanosaeta* sp.), the *Methanosarcina* sp. in the R2 reactor are quite robust toward different impairments. These include high tolerance to ammonium nitrogen concentration, salts, pH shocks, and high acetate concentrations [64,65]. The overall average performances of the R1 and R2 UASB reactors are listed in Table 3. The runs in both reactors were aimed to examine the effect of a change in OLR at a constant hydraulic retention time of 24 h. 

## 4. Conclusions

The results of the experiment showed that the R2 reactor successfully achieved high COD removal efficiency (>90%), biogas (27 L/d), methane (89%) production, and the specific methane yield of 0.21 LCH_4_/g COD_added_ at an organic loading rate (OLR) of 10 g L^−1^d^−1^ corresponding to 20 g COD/L at a hydraulic retention time (HRT) of one day, whereas the performance of the R1 reactor drastically declined at OLR 10 g L^−1^d^−1^ and in subsequent stages of the R1 reactor, especially the COD (48%), biogas (8 L), and methane composition (44%). 

The problems encountered in the R1 reactor at 10 g L^−1^d^−1^ significantly affected the performance of the reactor operation. This could be attributed to the shock load received by the system, the sludge washout which could be due to upflow velocity and short HRT of one day, and the insufficient development of microbial biomass (Methanosarcina bacteria) at higher OLR coupled with a significant accumulation of volatile fatty acids. Moreover, the absence of the attached growth material in the R1 reactor and a proper suspended solid settling material could also be the reason for the excellent performance of the R2 reactor against the R1 reactor. In addition, a comparison of the performance of the R2 reactor with literature studies for similar wastewater indicates that this study attained higher treatment efficiencies as compared with others at higher loading rates. The high COD removal efficiency of the R2 reactor is an additional advantage to scale up the system for energy production to a commercial scale in the slaughterhouse. However, post-treatment of the effluent was found necessary to achieve land or inland water discharge standard, especially during the treatment of CSWW at OLR 14 and 16 g L^−1^d^−1^, corresponding to 28,000, and 32,000 mg COD/L.

Additionally, several recommendations are listed for future work to improve the knowledge of the related study area.
More investigations should be carried out on similar wastewater with high lipid content (palm oil mill effluent) in order to investigate the capability and efficiency of the system;Similar studies should be performed on thermophilic condition in order to examine the rate of biodegradability under varied OLR and HRT.

## Figures and Tables

**Figure 1 ijerph-17-00283-f001:**
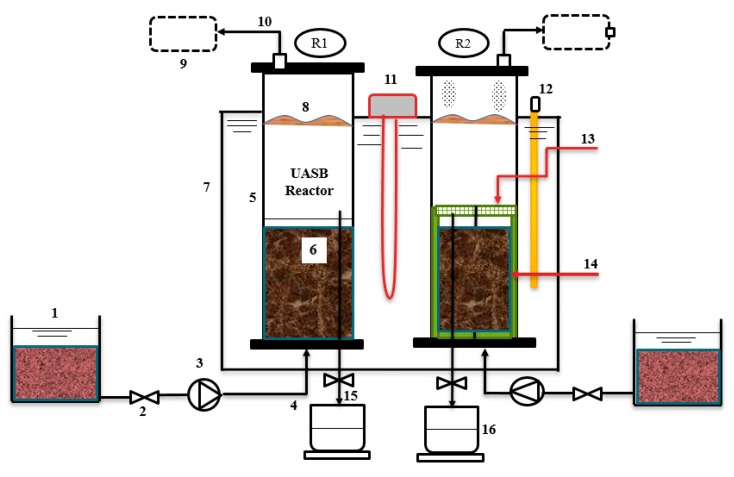
Schematic diagram of the upflow anaerobic sludge blanket (UASB) reactors sector: (**1**) Substrate, (**2**) open and close valve, (**3**) dosing pump, (**4**) influent, (**5**) PVC UASB reactor, (**6**) sludge, (**7**) water bath, (**8**) substrate level, (**9**) Tedlar bag, (**10**) biogas outlet, (**11**) heater, (**12**) thermometer, (**13**) PVC mesh coupled synthetic grass, (**14**) synthetic grass as attached growth, (**15**) effluent outlet, and (**16**) outlet container.

**Figure 2 ijerph-17-00283-f002:**
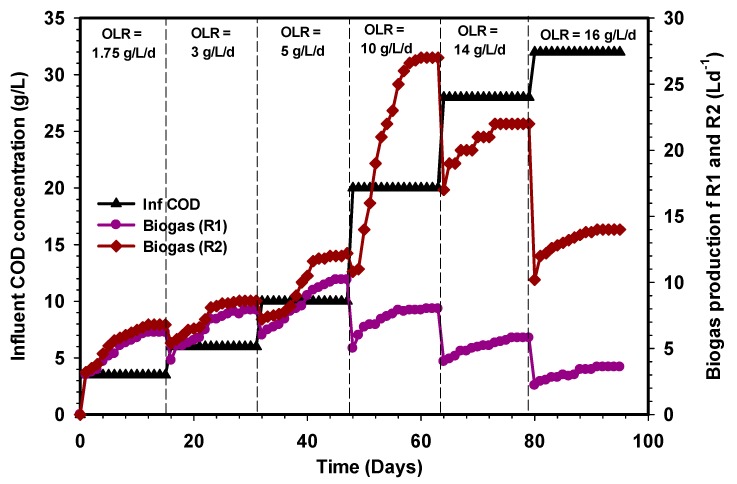
Daily biogas production during the period of operation of the conventional (R1) reactor and the modified (R2) reactor.

**Figure 3 ijerph-17-00283-f003:**
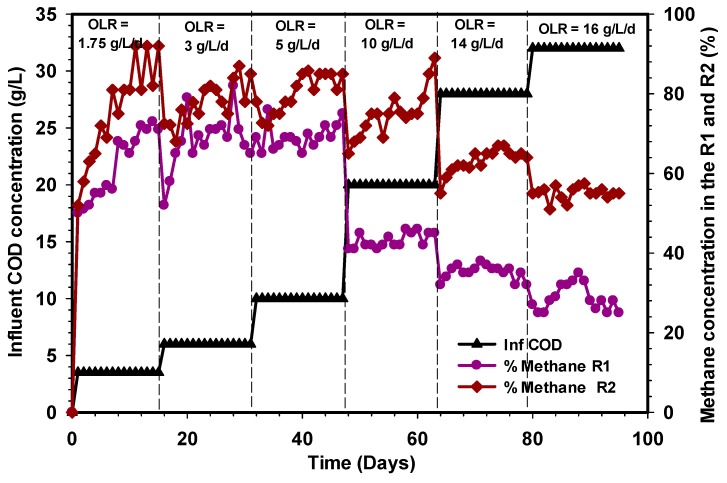
Composition of methane during the period of operation of R1 and R2 reactors.

**Figure 4 ijerph-17-00283-f004:**
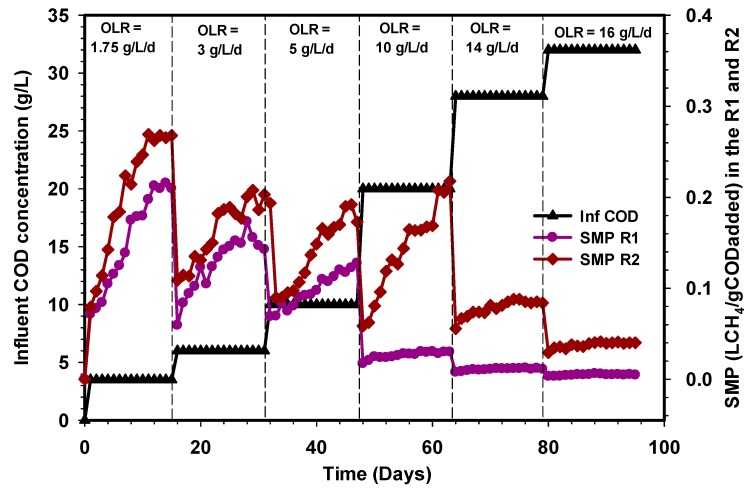
Variation of specific methane production (SMP) in the R1 and R2 reactors.

**Figure 5 ijerph-17-00283-f005:**
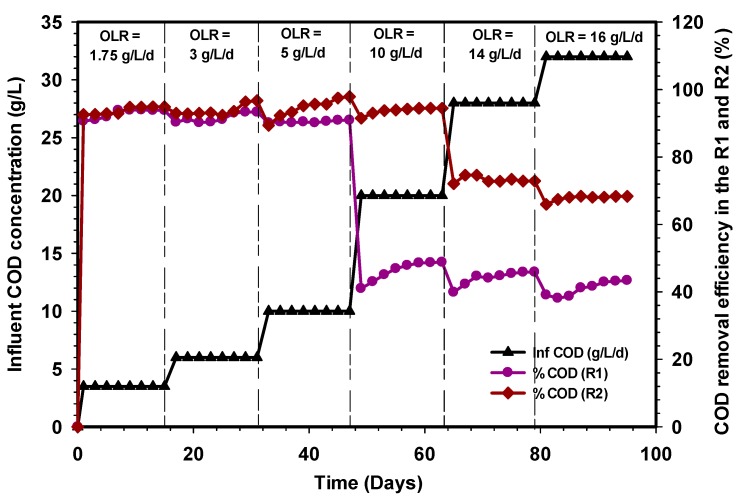
Variation in the COD removal efficiency of the R1 and R2 reactors.

**Figure 6 ijerph-17-00283-f006:**
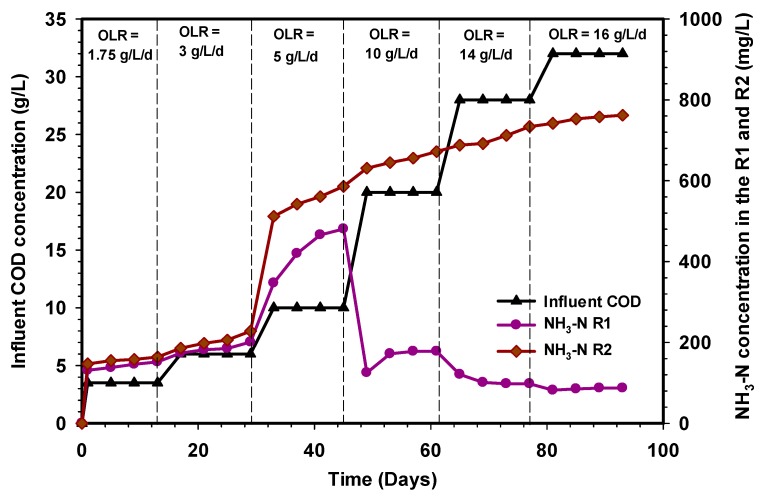
Variation of ammonia nitrogen concentration (NH_3_-N) in R1 and R2 reactors.

**Figure 7 ijerph-17-00283-f007:**
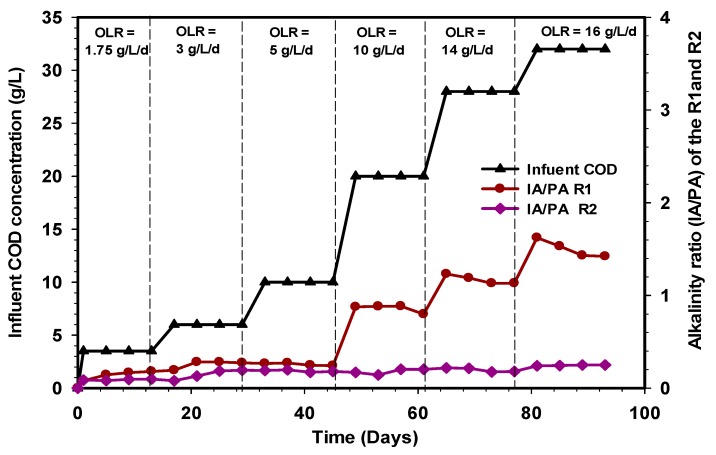
Variation of alkalinity ratio in the R1 and R2 reactors.

**Figure 8 ijerph-17-00283-f008:**
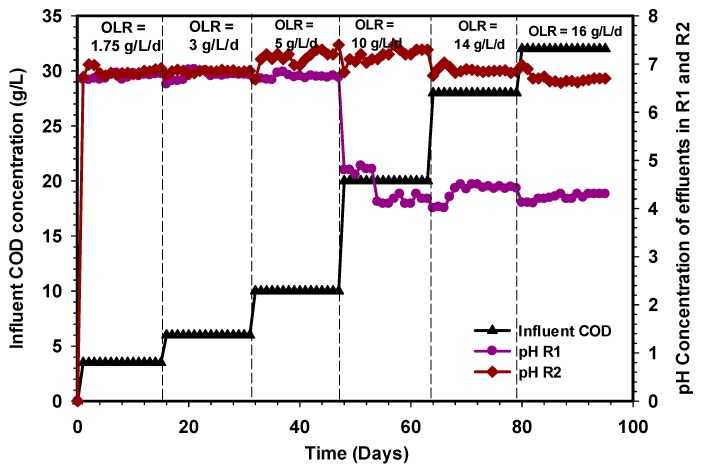
pH profile of the R1 and R2 reactors.

**Figure 9 ijerph-17-00283-f009:**
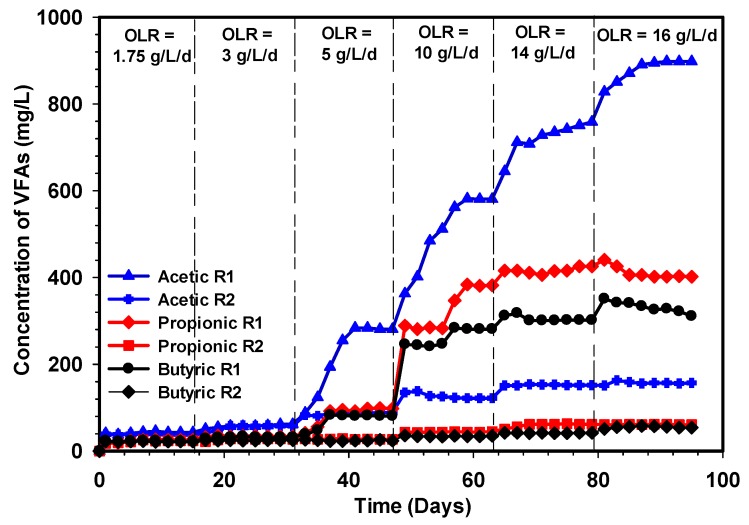
Variation in the VFA concentration for the R1 and R2 reactors.

**Figure 10 ijerph-17-00283-f010:**
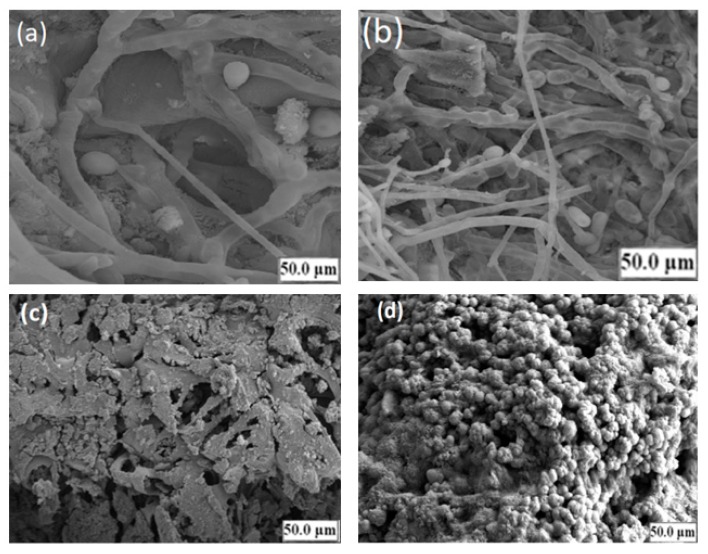
Represent the morphologies of inoculum in the R1 reactor (**a**) and the R2 reactor (**b**) after acclimatization (**c**,**d**) after contact with CSWW, (**a**,**b**) showing *Methanosaeta* sp. like morphology and (**c**,**d**) depicts *Methanosarcina* sp. like morphology.

**Table 1 ijerph-17-00283-t001:** Characteristics of the cattle slaughterhouse wastewater (CSWW).

Parameters	Unit	Average Value
pH	-	6.9 ± 0.8
Temperature	°C	27.6 ± 0.5
COD	mg/L	32,000 ± 112
BOD_5_	mg/L	17,158 ± 95
TSS	mg/L	22,300 ± 212
VSS	mg/L	18,924 ± 318
FOG	mg/L	1024 ± 34
TN	mg/L	1865 ± 18
PO_4_^3−^	mg/L	49 ± 2
Protein	mg/L	11,637.6 ± 18
NH_3_-N	mg/L	38 ± 1
Alkalinity	mg/L as CaCO_3_	582 ± 14
Color	Pt-Co	16,426.8 ± 334
Turbidity	FAU	12,500 ± 76

**Table 2 ijerph-17-00283-t002:** Startup strategy of conventional and the modified UASB reactors.

Fixed Parameters			Units		Value
Temperature			°C		36 ± 1
HRT			d		1
Experimental run	Day	Feed COD concentration (g L^−1^)	Corresponding OLR (g L^−1^d^−1^)	Dilution factor	Influent volume/flow rate
Stage I	1–15	3.5	1.75	9.2	6 L/d
Stage II	15–29	6.0	3	5.4	6 L/d
Stage III	29–47	10	5	3.2	6 L/d
Stage IV	47–63	20	10	1.6	6 L/d
Stage V	63–79	28	14	1.2	6 L/d
Stage VI	79–95	32	16	1	6 L/d

**Table 3 ijerph-17-00283-t003:** Summary of the average steady-state performance comparative study of the conventional (R1) and modified (R2) UASB reactors effluent and biogas production.

Fixed Parameter																			
HRT 24 h																			
Run	Duration (Days)	Influent COD (g L^−1^)	OLR (g L^−1^ d^−1^)	Biogas Production (L/d)		Methane Content (%)		SMP (LCH_4_/gCOD _added_)		pH		Alkalinity (mg/L) R1		Alkalinity (mg/L) R2		Alkalinity Ratio (IA/PA)		NH_3_-N (mg/L) Effluent	
		R1,R2	R1,R2	R1	R2	R1	R2	R1	R2	R1	R2	IA	PA	IA	PA	R1	R2	R1	R2
I.	15	3.5	1.75	6.2	6.8	71	88	0.210	0.28	6.6	6.8	42	302	32	322	0.14	0.09	141	156
II.	16	6.0	3	7.9	8.6	67	83	0.150	0.19	6.7	6.8	43	306	42	219	0.14	0.19	185	204
III.	17	10.0	5	10.2	12.2	72	83	0.120	0.18	6.7	7.3	45	248	41	235	0.18	0.17	428	550
IV.	16	20.0	10	8.0	27.0	44	89	0.020	0.21	4.2	7.3	51	58	35	175	0.88	0.20	163	651
V.	16	28.0	14	5.8	22	33	64	0.010	0.08	4.4	6.8	68	61	32	185	1.11	0.17	104	712
VI.	15	32.0	16	3.6	14	26	55	0.004	0.04	43	6.7	72	51	37	154	1.42	0.24	1590	753

COD: chemical oxygen demand; HRT: hydraulic retention time; SMP: specific methane production; NH_3_-N: ammonia nitrogen; IA: intermediate alkalinity; PA: partial alkalinity.

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
