# Peer review of "A Comparative Study of Biogas Production from Cattle Slaughterhouse Wastewater Using Conventional and Modified Upflow Anaerobic Sludge Blanket (UASB) Reactors"

_ijerph, 2019, doi:10.3390/ijerph17010283_

Round 1
Reviewer 1 Report
First of all, congratulations to the authors for the original and interesting research.
Comments and Suggestions:
Introduction:
Please, complete the introduction with more recent references. First ones are from 5-7 years ago.
Lines 45-47: Microorganisms are the only problem with waste waters (WW)? Please, complete the sentence and explain better. Complete it with a reference about the information given. Lines 65-67: Please, improve this sentence. In this way its kind of meaning that nitrogen is directly disposed to the environment. Relate it with WW emissions that due to the high nitrogen content can lead into eutrophication. Line 73: What legal restrictions? Mention them and its reference (example https://doi.org/10.3390/app8101804) Line 77: Please, indicate the volume reduction and average data of biogas produced. (Reference: https://doi.org/10.3390/app8101804) Lines 82-83: Please, indicate the approximate % of each gas with its reference.Materials and Methods:
When talking about the APHA Methods, please indicate its number (ie. Solids 5250-G). It could be a good idea to include in Table 1 a column with the method code and reference.
Table 1: Protein content is important due to its relation with the organic nitrogen. However, lipid and carbohydrate content are also important as they an determine the development of the anaerobic process. Please, include those measures.It is not necessary to mention the brand of the pH-meter or other fungible materials.
Lines 150-158: As the APHA Method code has been mentioned previously, this part can be resumed, just mentioning the method (i.e. it is not necessary to indicate the complete method name, partial alkalinity by tritation... please, only mention the method code). Line 164 and ...: Please don mention Agilent Technologies, Santa Clara USA. It is only necessary to mention the brand before the equipment (i.e gas chromatograpg Agilent HP 6890 N).Results and discussion:
Please, use colours ot other plotting dots (not squares or triangles because they are difficult to differentiate) when plotting the graphs.
Figures ,6,7 and 9: Include a pH graph, as it is a clear indicator of the process development. Figure 8: pH concentration? Just write pH Please, indicate the fixing method for the SEM (HDMS, Osmotic dehydration with Ethanol, critical point drying), and if the sample needs to be covered in Au, Pt... Table 3: Influent COD (gL-1) Liters of? influent, digestate, sludge? Biogas production? The volume is measured in normal conditions? Please indicate it in NL. Commonly, biogas production is expressed in gross production (Lbiogas/kg_influent) or specific production (Lbiogas/g_VS_influent) Please, indicate both measures. Methane production (%): That is not methane production, that's the composition of the produced biogas. Change the name of the column and add a new column with the methane production (LCH4/kg_influent and LCH4/g of VS influent). Column SMP: include it in the methane production column under a subtitle like, methane production related to added COD. It is usually measured as CH4/COD_removed, and it is also more rigorous. Add another column refering the measure to COD_removed. Alkalinity is usually used as an indicator of VFA formation. Please. split the alkalinity ratio column into three columns. One for IA, another for PA, and the last one for the ratio. Ammonia column: It is measured in mg/L. Liters of? influent, digestate... pH column should go before Alkalinity and Ammonia, as it is the most used indicator.Conclusions:
Please, write the conclusions, not in a single paragraph, use instead a bullet list.
Lines 391-393 can be avoided. They are redundant. Just indicate in a single bullet the modifications carried ut in R2.
Other suggestions:
Please, include a list of acronyms.
Author Response
Good day
On behalf of all the authors, we thank you most sincerely for the wonderful review of our manuscript and knowledge impacting comments. We have uploaded the respond to the comments raised for your kind consideration.
Thank you

Reviewer 2 Report
The overall manuscript is organized well. Clearly defined the experimental approach. The major criticism from this reviewer is the lack of cohesion in the conclusion section. The authors state the major findings in their comparative study of a conventional UASB (R1) and a modified USAB reactor (R2) in the ability to effectively remove COD, decrease biogas and methane production. The authors state the results but do not come full circle to suggest what the environmental implications are of use of the R2 reactor. While more investigation of wastewater with a higher lipid content the authors should describe the overall merits of their findings are it relates back to overall lower emissions and more efficient COD removal. Only minor edits are required to the format and organization of the manuscript.
Abstract - consistency of mesophilic temp - either use 35 (line 22) or 37 (line 130) or range 35 - 37.
Introduction -
Line 42 - noun-verb agreement - run on sentence line 46 - authors use "amount of hazardous" and pathogenic - should use "potentially hazardous and potentially pathogenic - no evidence to suggest otherwise line 48 - can you make comparison between m3 and percentage? line 65 - how does color contribute to environmental impact - it is unclear Objective in Introduction - clear and conciseM&M -
Table 1 - clearly defines characteristics of wastewater Table 2 - clearly shows the starting composition of USAB reactorsResults and Discussion -
Title for figure 5 - unclear line 339 - English - rephrase last two sentences of paragraph line 351 - Do authors know the scum found is cause of poor performance - any evidence to support or are they surmising that this is the cause. Line 372 - long filaments suggest presence of this organism unless they performed definitive tests - ie DNA sequence analysis, etc. When was this scanning EM analysis performed - day 95? rod-shaped not Rod-shaped Figure 10 are c and d from the reactors or images acquired elsewhere it is unclear by the use of "like images" in the figure legend. Authors need to clarify that these images are isolated samples from the reactors.Conclusions -
Extrapolate what the high COD removal means in bigger environmental picture. put the recommendations in paragraph form - 2 are listed
Author Response
Good day
On behalf of all the authors, I thank you most sincerely for the wonderful review, suggestions and overall, the knowledge impacting comments you made on our manuscript. We have uploaded the respond to the comments raised for your kind consideration.
Thank you

Round 2
Reviewer 1 Report
INTRODUCTION
Introduction still needs to be completed with updated references such as:
- https://doi.org/10.3390/app8101804
- https://doi.org/10.3390/en12173244
- https://doi.org/10.1080/17597269.2018.1506277
- https://doi.org/10.3390/app8112083
Line 73: What legal restrictions? Please explain the legal restrictions, mention them and ther references (laws, directives, council communications…)
MATERIALS AND METHODS.
Please, indicate the APHA Method Code (i.e. 2450-G) followed for the experiments.
Line 164 and ...: Please don´t mention Agilent Technologies, Santa Clara USA. It is only necessary to mention the brand before the equipment (i.e gas chromatograpg Agilent HP 6890N).
RESULTS AND DISCUSSION
Please, include a pH plot in every graph.
Daily biogas production is reported in liters, however it is neccersary to indicate pressure and temperature, as gas volumes vary. That´s why I suggested to express it in Nl. Please. correct the measures in terms of pressure and temperature and express it in Nl.
Ammonia column: It is measured in mg/L. Liters of? influent, digestate...
Author Response
Thank you very much for the wonderful and knowledge impacting comments and review. Please find attached a copy of the response to the comments and the revised manuscript for your kind consideration.
